# Cytotoxicity Effect of Constituents of *Pinus taiwanensis* Hayata Twigs on B16-F10 Melanoma Cells

**DOI:** 10.3390/molecules27092731

**Published:** 2022-04-23

**Authors:** Man-Hsiu Chu, Sui-Wen Hsiao, Yu-Chen Kao, Hwa-Wen Yin, Yueh-Hsiung Kuo, Ching-Kuo Lee

**Affiliations:** 1Graduate Institute of Pharmacognosy, Taipei Medical University, 250 Wu Xin Street, Taipei 11031, Taiwan; m303110003@tmu.edu.tw; 2Ph.D. Program in Drug Discovery and Development Industry, College of Pharmacy, Taipei Medical University, 250 Wu Xin Street, Taipei 110301, Taiwan; d343106004@tmu.edu.tw; 3Graduate Institute of Pharmacy, Taipei Medical University, 250 Wu Xin Street, Taipei 11031, Taiwan; yckao1225@gmail.com; 4Forestry Research Institute, Council of Agriculture, Executive Yuan, 53, Nanhai Road, Taipei 10066, Taiwan; hawnyin@gmail.com; 5Department of Chinese Pharmaceutical Sciences and Chinese Medicine Resources, College of Pharmacy, China Medical University, Taichung 40402, Taiwan; kuoyh@mail.cmu.edu.tw; 6Department of Biotechnology, Asia University, Taichung 41354, Taiwan; 7Department of Chemistry, Chung Yuan Christian University, Zhongbei Road, Zhongli District, Taoyuan 32023, Taiwan

**Keywords:** *Pinus taiwanensis* Hayata, *Pinaceae*, B16-F10 melanoma cells, UHPLC-MS/MS

## Abstract

*Pinus taiwanensis* Hayata (*Pinaceae*) is an endemic plant in Taiwan. According to the Chinese Materia Medica Grand Dictionary, the *Pinus* species is mainly used to relieve pain, and eliminate pus and toxicity. In this study, nineteen compounds were isolated from the ethyl acetate layer of the ethanolic extract of *P. taiwanensis* Hayata twigs using bioassay-guided fractionation, and their anti-melanoma effects were investigated through a B16-F10 mouse melanoma cell model. The structures of the purified compounds were identified by 2D-NMR, MS, and IR, including 1 triterpenoid, 9 diterpenoids, 2 lignans, 4 phenolics, 1 phenylpropanoid, 1 flavonoid, and 1 steroid. Among them, compound **3** was found to be a new diterpene. Some of the compounds (**2**, **5**, **6**, **17**, **18**) showed moderate cytotoxicity effects. On the other hand, the anti-melanoma effect was no better than that from the original ethyl acetate layer. We presumed it resulted from the synergistic effect, although further experimentation needs to be performed.

## 1. Introduction

Melanoma is initially formed from a malignant transformation of melanocytes in the basal layer of epidermal cells. When the skin is exposed to the sun for a long time, the melanocytes continue to proliferate and accumulate, forming the common nevus on the skin. It is usually a benign nevus at first. However, a nevus that abnormally darkens in color or changes in shape and size is the precursor of melanoma. Current known risk factors for melanoma include long-term exposure to ultraviolet radiation, structural and quantitative moles, genetic mutations, familial inheritance, past skin cancer, radiation therapy, chemical carcinogens, and immunodeficiency [1]. UV radiation is an established risk factor for melanoma. As people become more frequently exposed to natural and/or artificial UV radiation, public concerns of melanoma rise [2]. The median age of people diagnosed with melanoma is 52 years old, which is younger than most solid tumors [3]. There are 22.8 melanomas per 100,000 people per year. In 2021, there was an estimate of 106,110 diagnoses, and 7180 deaths from melanoma [4]. Characteristics of metastasis melanoma include asymmetry, a blue-white veil, structureless areas, and atypical vascular structures [5]. Compared to other cancers, melanoma is fairly common [4].

*Pinus taiwanensis* Hayata belongs to *Pinaceae.* It is endemic to Taiwan, and is widely distributed in the mountainous area at an altitude of 700–3500 m. It is a large evergreen tree with a straight trunk of length up to 35 m and a diameter of 80 cm. The bark is gray-brown, longitudinally grooved, and irregularly flaky. The twigs grow horizontally. The dark green leaves are rigid in quality with two needles in a bunch, each 8 to 11 cm long and often possessing 4 to 7 fat grooves [6]. Chinese ancient writing says that *Pinus* was the top-grade drug in *Shen Nong Ben Cao Jing*. However, as far as we know, the research publications for *P. taiwanensis* Hayata are still limited. The biological activities of *P. taiwanensis* Hayata include anti-inflammatory [7], antioxidant [8], anti-lipemic [9], anti-glycemic [9], and antitumor effects [10,11]. The major components found are terpenoids [12], steroids [13], flavonoids [7], and alkanes [13]. In the preliminary stage of this study, we used ultra-high-performance liquid chromatography tandem high-resolution mass spectrometry (UHPLC-MS/MS) to analyze the signals of the twig extracts of *P. taiwanensis* Hayata, and found thousands of mass-to-charge ratios (*m/z*) that had never been reported previously [7,13]. These observations indicate that there might be lots of unknown components worthy of exploration. Moreover, the crude extract of *P. taiwanensis* Hayata showed obvious cytotoxic effects on B16-F10 melanoma cells. Therefore, the aim of this study is to isolate the potential novel components from the twigs of *P. taiwanensis* Hayata, and to evaluate their cytotoxicity against a melanoma B16-F10 cell model.

## 2. Results

### 2.1. Bioassay-Guided Compound Isolation from Pinus taiwanensis Hayata Twigs

The twigs of *P. taiwanensis* Hayata were soaked in ethanol at a ten times volume-to-weight ratio, extracted three times, and concentrated to obtain a crude extract. The crude extract of twigs of *P. taiwanensis* Hayata (CE) was analyzed through UHPLC-MS/MS. As shown in the chromatogram (Figure 1), thousands of *m/z* were never reported in the studies of *P. taiwanensis* Hayata. The crude extract was subsequently partitioned to obtain the ethyl acetate (EA) layer, *n*-butanol (BU) layer, and water (H) layer of the extract.

We performed the analysis on the B16-F10 cells and the results showed that the EA layer was the most effective layer (data no shown). Then, we continued with the MTT assay. The results of the cytotoxicity test of crude extracts on B16-F10 cells are shown in Figure 2. The CE of *P. taiwanensis* Hayata at 50 μg/mL significantly reduced the cell survival rate to 23.18% ± 10.85% (mean ± standard deviation). Moreover, the extract of the EA layer was the most cytotoxic among three partitions, with a cell survival rate of 14.10% ± 1.95% at the concentration of 50 μg/mL. Based on the obvious cytotoxic effect shown in the EA layer, it was selected for further analysis.

### 2.2. Compounds Isolated from the Ethyl Acetate Layer of Pinus Taiwanensis Hayata Twigs

The EA layer was further subjected to fractionation and compound isolation. As a result, 19 compounds were isolated and identified by 2D NMR, MS, and IR, then compared to spectroscopic data reported in the literature. Among them, compound **3** was identified as a new compound (Figure 3). The structural and chemical names of the isolated compounds are listed in Table 1, and the structures of compounds are shown in Figure 4. There was: one triterpenoid, 3β-methoxyserrat-14-en-21-one (**1**) [14,15]; nine diterpenoids, dehydroabietic acid (**2**) [16], 9α,13β-dihydroxyabiet-8 (14)-enoic acid (**3**), 7-oxodehydroabietic acid (**4**) [17], 8α-podocarp-13-en-15-oic acid (**5**) [18], 12α-hydroxyabietic acid (**6**) [19], 15-hydroxyabietic acid (**7**) [20], 15-hydroxydehydroabietic acid (**8**) [21,22], 7β-hydroxydehydroabietic acid (**9**) [23], and 15-hydroxy-7-oxo-8,11,13- abietatrien-18-oic acid (**10**) [24]; two lignans, mataresinol (**11**) [25] and pinoresinol (**12**) [26]; four phenolics, 4-hydroxy-benzaldehyde (**13**) [27], vanillin (**14**) [28], vanillic acid (**15**) [29], and fumalic acid (**16**) [30]; one phenylpropanoid, caffeic acid (**17**) [31]; one flavonoid, taxifolin (**18**) [32]; one steroid, β-sitosterol (**19**) [33].

#### Structure Analysis of **3**

Compound **3** was a yellow oil. The data of the high-resolution electrospray ionization-tandem mass spectrometry exhibited a deprotonated molecular ion [M−H]^−^ at *m/z* 335.2218 (cal. 335.2217) in negative ion mode. Together with the ^13^C-NMR data, the molecular formula of compound **3** was speculated to be C_20_H_32_O_4_.

According to the DEPT-NMR spectrum, there were four methyl groups, seven methylene groups, three methine groups, and six quaternary carbons. The ^1^H-NMR spectrum of **3** (Appendix A) showed signals for methyl groups at δ_H_ 0.94 (d, *J* = 6.8 Hz), 0.95 (d, *J* = 6.8 Hz). Since both were related to the δ_H_ 1.88 (1H, septet) signal from the ^1^H–^1^H COSY spectrum (Appendix A), it could be deduced as an isopropyl group. δ_H_ 1.06 (s) and 1.26 (s) were peaks of methyl groups, in the ^13^C-NMR spectrum (Appendix A), and δ_C_ 17.5 and 19.1 were presumed to have two absorption signals connected to the methyl group on the quaternary carbons, and an olefinic proton at δ_H_ 6.07 (s) (Table 2), presumably confirmed by ^13^C-NMR and DEPT experiments at δ_C_ 144.1 (4° carbon) and 127.0 (CH) as the double-bond absorption signal of the group. The ^13^C-NMR spectrum at δ_C_ 184.0 indicated that **3** had a carboxylic acid group. For compound **3**, the degree of unsaturation was 5. After subtracting one carboxylic acid and one double bond, the remaining degree of unsaturation was 3. Inferred to be three six-ring structures, it was speculated that compound **3** possesses an abietane skeleton in the diterpenoids.

The ^13^C-NMR spectrum showed two quaternary carbons of δ_C_ 79.1 and 80.7, which may be affected by the hydroxy group and appeared in the lower field (downfield) position. The HMBC experiment (Figure 5, Appendix A) showed that δ_H_ 0.95 was related to δ_C_ 79.1 (C-13), δ_C_ 32.1 (C-15), and δ_C_ 17.1 (C-17), δ_H_ 0.94 was related to δ_C_ 79.1 (C-13), and δ_H_ 1.88 and 6.07 were related to δ_C_ 79.1 (C-13). In conclusion, isopropyl was located on C-13 and δ_C_ 79.1 was in the C-13 position. Furthermore, the relationship between δ_H_ 2.41, 2.50, 2.05, 6.07, 1.06, and δ_C_ 80.7 (C-9) suggests that the position of δ_C_ 80.7 is C-9. According to the results of HMBC experiments, δ_H_ 6.07 was related to δ_C_ 79.1 (C-13), 80.7 (C-9), 32.1 (C-15), and 24.2 (C-7), which confirmed that the double bond of δ_C_ 144.1 and 127.0 is positioned at C-8 and C-14.

As for the stereochemistry, it could be explained by the NOESY spectrum (Figure 5, Appendix A). There was NOESY association between H-20 and H-19, which showed that -COOH was located in the equatorial direction. H-20 was related to H-11, which meant that 9-OH was a pseudo-axial. H-12 was related to H-17, which indicated that 13-OH was located on the axial.

Based on the above information, compound **3** was identified as a new compound named 9α-13β-Dihydroxyabiet-8(14)-enoic acid, that had not been documented in the literature. The structure is shown in Figure 4.

### 2.3. Evaluation of Cytotoxicity

The toxicity effect of pure compounds **1**–**19** (except **3**) isolated from *P. taiwanensis* Hayata twigs on B16-F10 cells is shown in Figure 6. At a concentration of 50 μM, compounds **2**, **5**, **6**, **17**, and **18** showed a statistic cytotoxic effect. In order to explore its possible causes, the compounds were once again mixed in the ratio of weights of the purified compounds. Compounds **6**, **7**, **8**, **9**, and **11** were mixed back as fraction 4′, while compounds **10**, **12**, **15**, **16**, **17**, and **18** were mixed back as fraction 5′. The cell survival rates at different concentrations of fraction 4′ were 24.77% at 100 μg/mL, 42.40% at 50 μg/mL, and 82.57% at 25 μg/mL. The cell survival rates of fraction 5′ were 12.38% at 100 μg/mL, 21.20% at 50 μg/mL, and 46.64% at 25 μg/mL. The results showed that fraction 4′ and fraction 5′ had more cytotoxic effects than pure compounds. The cytotoxic effects on B16-F10 melanoma cells were increase when remixed, which could be attributed to the synergistic effect.

## 3. Discussion

The study of secondary metabolism on *P. taiwanensis* Hayata is rare, and therefore it is necessary to distinguish this plant from the site of phytochemistry. Moreover, the great cytotoxicity on B16-F10 cells made *P. taiwanensis* Hayata more valuable to discuss. This was the first metabolites research focusing on the twigs of the *P. taiwanensis* Hayata. As a result, 19 compounds were isolated, including a new diterpenoid. Moreover, 14 of these 19 compounds had not been found in other parts of *P. taiwanensis* Hayata. The extracts of twigs of *P. taiwanensis* Hayata were separated and purified before 19 compounds were obtained in the EA layer. Structures of these compounds included 1 triterpenoid, 9 diterpenoids, 2 lignans, 4 phenolics, 1 phenylpropanoid, 1 flavonoid, and 1 steroid [34]. Diterpene was the main component, and it showed the same tendency as other *Pinus* plants. The compounds l*-α*-pinene, dl-limonene, bornyl acetate, and d-longifolene had previously been found in *P. taiwanensis* Hayata [12]. Among them, 9α,13β-dihydroxyabiet-8(14)-enoic acid (**3**) was found as a new compound. The above results showed that the low-polarity part of the composition of *P. taiwanensis* Hayata twigs was dominated by diterpenoids. This result provided an understanding of the composition of secondary metabolites of *P. taiwanensis* Hayata twigs.

According to the previous studies, less than 100 compounds were identified in all parts of *P. taiwanensis* Hayata, and 70 compounds matched the *m/z* signals detected in UHPLC-MS/MS analyses. On the other hand, the ion signal of a few numbers of substances had not been found in the UHPLC-MS/MS chromatogram, which may be absent or minor in the *P. taiwanensis* Hayata twigs. This was confirmed with the 19 main compounds isolated from the twigs of *P. taiwanensis* Hayata, and only 5 components were repeated from the previous research [7]. The 19 main substances isolated from the *P. taiwanensis* Hayata twigs were detected by UHPLC-MS/MS as well, but the rest of the thousands of *m/z* signals represented that most of the compounds were not explored yet, which are worthy of future study. UHPLC-MS/MS is suitable for the exploration of plant metabolites, and could be an important tool to investigate the composition of natural products.

In the diterpenoids **2** to **10** from *P. taiwanensis* Hayata, the difference in isopropyl chemical shifts on C-13 can be found, and they were grouped into three types. Abietane type (compounds **2**–**9**) could determine the attached functional group by the H-15 isopropyl group. If the isopropyl group was connected to the benzene ring next to it, the chemical shift of 15-H of the isopropyl group was about 2.8 ppm (e.g., compound **2**, dehydroabietic acid). If the isopropyl group was attached with a double bond, the 15-H hydrogen spectrum shift of the isopropyl group was 2.4 ppm (e.g., compound **6**, 12α-hydroxyabietic acid), and if the isopropyl group was attached with -OH, its chemical shift was 1.8 ppm (e.g., compound **3**, 9α-13β-dihydroxyabiet-8(14)-enoic acid). The literature corroborated this phenomenon [35].

Due to the richness and diversity of natural plant compounds, natural products and their derivatives account for more than half of the development of anti-cancer drugs. In this experiment, *P. taiwanensis* Hayata, which is unique to Taiwan, was used as the research subject. Despite the results showing that the pure isolated compounds were not as effective as pre-existing drugs, the toxic effects on B16-F10 melanoma cells were retained when remixed during a cytotoxicity test, which could be attributed to the synergistic effect. The toxicity of the crude extract on B16-F10 melanoma cells is worthy of further investigation.

Regarding the uniqueness of this Taiwan endemic species, more activity screening tests can be performed on the pure compounds isolated from this plant in hopes that more special structures and mechanisms of action can be purified and isolated in the future.

## 4. Materials and Methods

### 4.1. Chemicals and Reagents

Penicillin-streptomycin solution (PS), trypsin-EDTA solution, dimethyl sulfoxide (DMSO), thiazolyl blue tetrazolium bromide (MTT), and Dulbecco’s Modified Eagle’s Medium-high glucose (DMEM) were purchased from Sigma-Aldrich (St. Louis, MO, USA). Fetal bovine serum (FBS) was purchased from SAFC Biosciences (Victoria, Australia). Methanol (ACS Grade), ethyl acetate (ACS Grade), dichloromethane (ACS Grade), and *n*-hexane (ACS Grade) were purchased from Mallinckrodt (St. Louis, MO, USA). *n*-Butanol (ACS Grade) was purchased from J. T. Baker. Methanol-*d*_4_ (CD_3_OD), acetone-*d*_6_ (CD_3_COCD_3_), and chloroform-*d* (CDCl_3_) were purchased from Merck (Darmstadt, Germany).

### 4.2. Plant Material

*Pinus taiwanensis* Hayata twigs were collected from the Forestry Research Institute in Taipei, Taiwan (coordinates N 25°1′52″; E 121°30′37″).

### 4.3. Extraction and Isolation

#### 4.3.1. Extraction

For extraction, 3.35 kg of dried *Pinus taiwanensis* Hayata twigs were extracted with 33.5 L of ethanol three times and evaporated in a rotary evaporator at 37 °C to obtain 230.02 g of crude extract. The crude extract was then suspended in water and partitioned with ethyl acetate and *n*-butanol. Thereafter, the ethyl acetate layer (170.19 g), the *n*-butanol layer (31.80 g), and the water layer (25.02 g) were obtained.

#### 4.3.2. Column Chromatography

The ethyl acetate layer (112.00 g) was chromatographed on silica gel and eluted stepwise with mixed solutions of *n*-hexane:ethyl acetate:methanol (10:0:0, 9:1:0, 8:2:0, 7:3:0, 6:4:0, 4:6:0, 2:8:0, 0:10:0, and 0:0:10 *v/v*) as mobile phases. A total of nine fractions were collected.

All compounds were obtained from Fr. 2, 3, 4, and 5 by normal phase, semi-preparative, high-performance liquid chromatography (HPLC) (Phenomenex^®^ Luna semi-preparative column: 250 × 10 mm), cooperating with an infrared radiation (IR) detector recorded on a JASCO FT/IR 4100 Spectrometer (Jasco, Tokyo, Japan) for detection. The flow rate was 3 mL/min and eluted with *n*-hexane and EA (Figure 3).

#### 4.3.3. NMR Analysis

NMR spectra were recorded at 25 °C using Bruker AV-300 MHz and AV-500 MHz spectrometers (Bruker, Rheinstetten, Germany). Chloroform-*d* was used as the internal lock. ^1^H–^1^H correlated spectroscopy (COSY), ^1^H–^1^H Nuclear Overhauser Effect Spectroscopy (NOESY), and heteronuclear multiple-bond correlation (HMBC) spectra were recorded.

#### 4.3.4. UHPLC-MS/MS Analysis

First, 1 mg of *P. taiwanensis* Hayata crude extract was dissolved in 1 mL of ethanol (ACS Grade). We then took the sample solution and diluted it 100 times with methanol (ACS Grade). The solution was filtered through a 0.22 μm filter to inject 1.0 μL for UHPLC-MS/MS analysis. Q Exactive Plus Hybrid Quadrupole Orbitrap Mass (Thermo Fisher Scientific Inc., USA) was used for qualitative and quantitative analyses to characterize the chemical composition of *P. taiwanensis* Hayata. It was subsequently analyzed on a Accucore TM Polar Premium column (150 × 2.1 mm, 2.6 µm) (Thermo Fisher Scientific Inc., Waltham, MA, USA) at 40 °C. The mobile phase was water (A) and 100% methanol (B) at a flow rate of 0.3 μL/min. The gradient was 0–1 min, 80% A; 1–25 min, 80–0% A; 25–30 min, 0% A; 30–30.1 min, 0–80% A; 30.1–35 min, 80% A, with an injection volume of 10 μL. Spray voltages were 3.5 kV in positive mode and −3.2 kV in negative mode, capillary temperature was 360 °C, source heater temperature was 350 °C, resolving power was 15,000, scan range was *m/z* 100–1000, higher-energy collisional dissociation (HCD) was 20, 30, 40, and 50 eV, and the process software was Xcalibur (version 2.2).

### 4.4. Cell Line and Culture

B16-F10 cells (ATCC number: CRL-6475) were cultured in DMEM supplemented with 10% FBS, penicillin (100 U/mL), and streptomycin (100 μg/mL) in a humidified 5% CO_2_ incubator at 37 °C.

### 4.5. Melanin Content Assay

The B16-F10 melanoma cells were seeded with 1 × 104 cells/well in 3 mL of medium in 6-well culture plates and incubated overnight. The cells were treated with different concentrations of the substance, compound, or fluorouracil for 72 h in the presence of 100 nM of α-MSH. At the end of the treatment, the cells were washed with PBS and lysed with 150 μL of 1 N NaOH (Merck, Darmstadt, Germany) containing 10% DMSO for 1 h at 80 °C. The absorbance was measured by a microplate reader at 405 nm. The melanin content of control cells was assigned as 100%, and the melanin content of compound-treated cells was calculated relative to the control group.

### 4.6. MTT Cell Viability Assay

The MTT assay was used to determine the cytotoxicity effects of the crude extract and compounds on B16-F10 cells. B16-F10 cells were treated with crude extracts or compounds for 48 h at 37 °C, then stained with MTT solution and further incubated for 2 h. Afterwards, supernatants were removed and 400 μL of DMSO was added to each well of the plate. The positive control was fluorouracil [36]. The absorbance of the solution was analyzed using a microplate reader (Molecular Devices, San Jose, CA, USA) under a wavelength of 570 nm. All experiments were carried out three times and the formula used to calculate the survival rate was as follows:(Absorbance value in the dosing place/Resting absorbance value) × 100%

### 4.7. Statistical Analyses

Statistical analysis of the data was performed with one-way analysis of variance (ANOVA), and Dunnett’s post-test was employed for multi-groups comparisons using GraphPad Prism 8 (GraphPad Software, Inc., San Diego, CA, USA). The results were expressed as mean ± SD of triplicates and differences were considered statistically significant when *p* < 0.05.

## 5. Conclusions

The constituents of *Pinus taiwanensis* Hayata twigs were diterpenoids, including a new compound, 9α,13β-dihydroxyabiet-8(14)-enoic acid (**3**). The cytotoxicity test of B16-F10 melanoma cells was performed on the 19 purified compounds. However, the isolated pure compounds showed limited cytotoxic effects, even lower than that of the crude extract at a concentration of 50 μM. Interestingly, the cytotoxicity was restored after we remixed the compounds, which may be due to potential synergistic effects. This will be explored in more detail in the future.

## Figures and Tables

**Figure 1 molecules-27-02731-f001:**
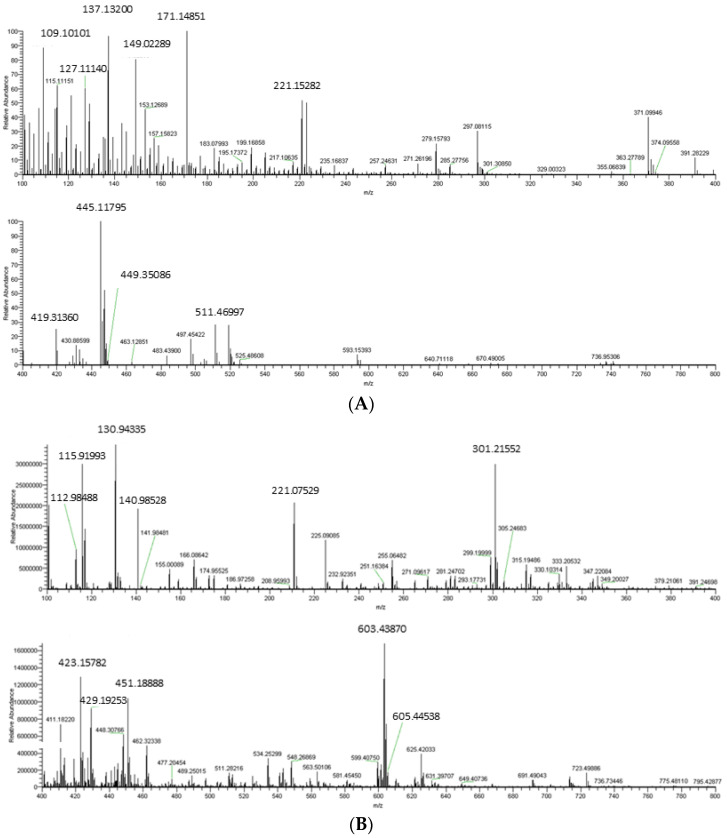
Mass spectrum of crude extract of twigs from *Pinus taiwanensis* Hayata. (**A**) Partial spectrum from 100 to 400 *m/z* and 400 to 800 *m/z*, in positive ion mode. (**B**) Partial spectrum from 100 to 400 *m/z* and 400 to 800 *m/z*, in negative ion mode.

**Figure 2 molecules-27-02731-f002:**
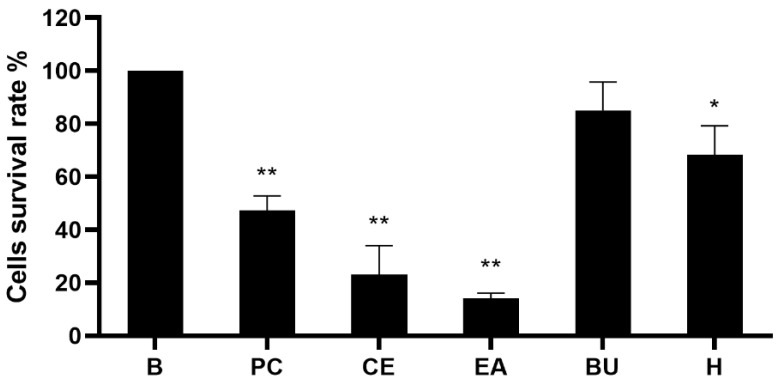
Cytotoxicity of crude extracts on B16-F10 melanoma cells. Cell toxicity of different parts of the extract were evaluated by the MTT assay on B16-10 cells. The EA layer, *n*-BuOH layer, and H_2_O layer were partitioned from the crude extract and tested in the concentration of 50 μg/mL. Positive control was 5 μM (0.65 μg/mL) of fluorouracil. Results were expressed as % of control cells and mean ± standard deviation (SD). The value was compared with different concentrations in each group (*n* = 3). B, control cell; PC, positive control; CE, crude extract; EA, EA layer; BU, *n*-BuOH layer; H, H_2_O layer. *p*-values were derived from one-way ANOVA with Dunnett’s multiple comparison tests. * *p*-value < 0.01, ** *p*-value < 0.001.

**Figure 3 molecules-27-02731-f003:**
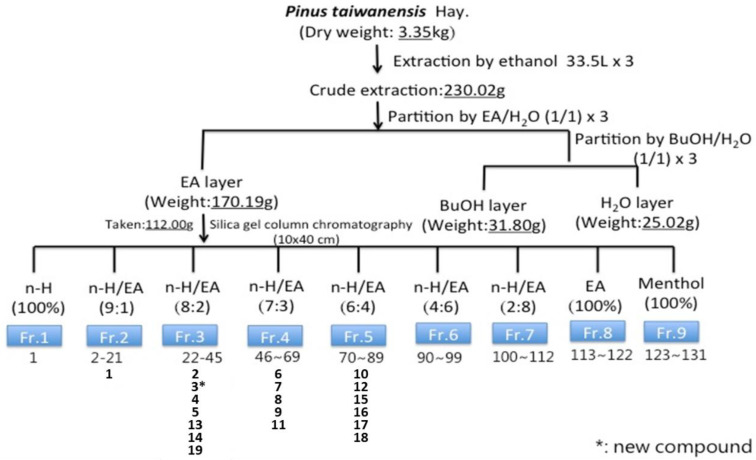
Flow diagram of bioassay-guided procedure. Bioassay-guided fractionation and isolation of the EA extract of *Pinus taiwanensis* Hayata twigs resulted in 19 isolated compounds. * is the new compound.

**Figure 4 molecules-27-02731-f004:**
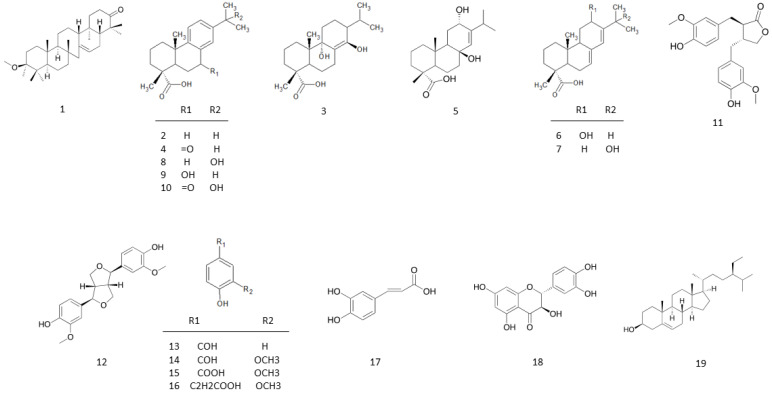
The structures of **1**–**19** from the twigs of *Pinus taiwanensis* Hayata.

**Figure 5 molecules-27-02731-f005:**
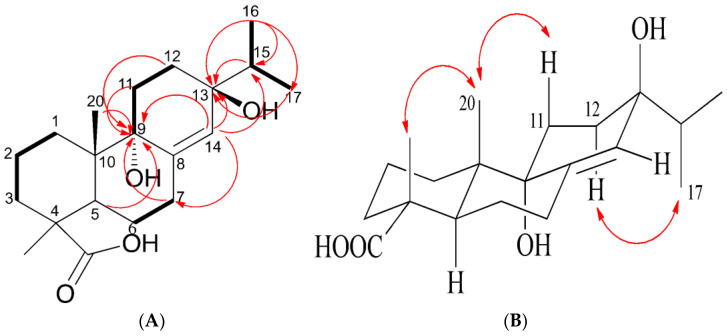
Selected 2D NMR (Chloroform-*d*) correlations for compound **3**. (**A**) COSY: **―**; HMBC: 
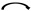
. (**B**) NOESY.

**Figure 6 molecules-27-02731-f006:**
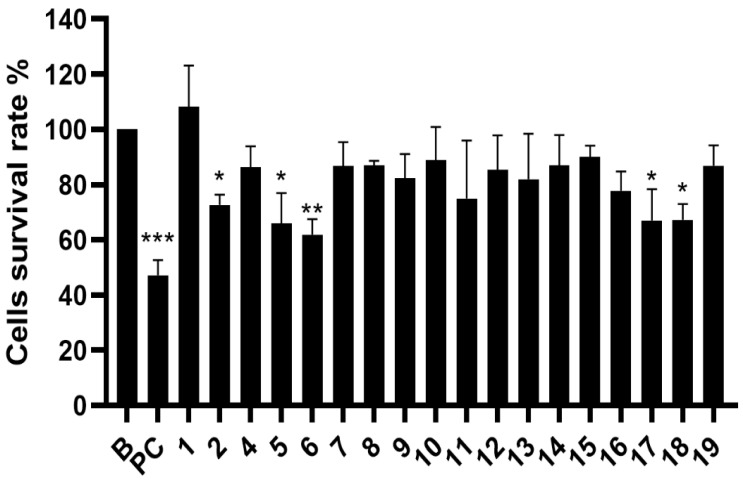
Cytotoxicity of pure compounds to B16-F10 melanoma cells. Results of cytotoxicity of each sample were expressed as % of control cells and mean ± SD (*n* = 3). B, control cell; PC, positive control (5-FU, 5 μM). Compounds 1–19 (50 μM), expect **3**. *p*-values were derived from one-way ANOVA with Dunnett’s multiple comparison tests. * *p*-value < 0.05, ** *p*-value < 0.01, *** *p*-value < 0.001.

**Table 1 molecules-27-02731-t001:** Compounds isolated from twigs of *Pinus taiwanensis* Hayata.

Chemical Structure Classification	Compounds Name
Triterpenoid	3β-Methoxyserrat-14-en-21-one (**1**)
Diterpenoid	Dehydroabietic acid (**2**)
	9α,13β-Dihydroxyabiet-8(14)-enoic acid (**3**)
	7-Oxodehydroabietic acid (**4**)
	8β-Podocarp-13-en-15-oic acid (**5**)
	12α-Hydroxyabietic acid (**6**)
	15-Hydroxyabietic acid (**7**)
	15-Hydroxydehydroabietic acid (**8**)
	7β-Hydroxydehydroabietic acid (**9**)
	15-Hydroxy-7-oxo-8,11,13-abietatrien-18-oic acid (**10**)
Lignan	Matairesinol (**11**)
	Pinoresinol (**12**)
Phenolic	4-Hydroxy-benzaldehyde (**13**)
	Vanillin (**14**)
	Vanillic acid (**15**)
	Fumalic acid (**16**)
Phenylpropanoid	Caffeic acid (**17**)
Flavonoid	Taxifolin (**18**)
Steroid	β-Sitosterol (**19**)

**Table 2 molecules-27-02731-t002:** ^13^C-NMR (125 MHz, Chloroform-*d*), ^1^H-NMR (500 MHz, Chloroform-*d*), and HMBC data for compound **3**.

Position	^13^C-NMR	^1^H-NMR	HMBC
	δc (Multiplet)	δ_H_ (Multiplet, *J* in Hz)	H→C
**1**	30.7	1.85 (m), 1.39 (m)	C-2, C-3, C-20
**2**	19.8	1.61 (m), 2.44 (m)	C-4, C-10
**3**	36.7	1.74 (m), 1.51 (m)	C-1, C-4, C-18, C-19
**4**	46.3	---	---
**5**	38.1	2.41 (m)	C-4, C-9, C-10, C-18, C-19, C-20
**6**	29.7	1.52 (m)	C-4, C-10
**7**	24.2	2.50 (m)	C-5, C-8, C-9, C-14
**8**	144.1	---	---
**9**	80.7	---	---
**10**	39.0	---	---
**11**	21.7	2.05 (m), 1.41 (m)	C-8, C-9, C-10, C-12
**12**	25.0	1.92 (m), 1.42 (m)	C-9, C-11, C-15
**13**	79.1	---	---
**14**	127.0	6.07 (s)	C-7, C-9, C-13, C-15
**15**	32.1	1.88 (septet, *J* = 6.8)	C-12, C-14, C-16, C-17
**16**	17.4	0.95 (d, *J* = 6.8)	C-13, C-15, C-17
**17**	17.1	0.94 (d, *J* = 6.8)	C-13, C-15, C-16
**18**	184.0(s)	---	---
**19**	17.5(q)	1.26 (s)	C-3, C-4, C-5, C-19
**20**	19.1(q)	1.06 (s)	C-1, C-5, C-9, C-10

## Data Availability

Not applicable.

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
