# Peer review of "Cytotoxicity Effect of Constituents of Pinus taiwanensis Hayata Twigs on B16-F10 Melanoma Cells"

_molecules, 2022, doi:10.3390/molecules27092731_

Round 1

Reviewer 1 Report

The manuscript entitled “Anti-Melanoma Effect of Constituents of Pinus taiwanensis Hayata Twigs on B16-F10 melanoma cells” showed an interesting subject in the secondary compounds field. The fractionation, purification, and characterization of secondary metabolites were carried out and the cytotoxicity of then were evaluated against melanoma cell line. The MTT assay is a very simple experiment to infer about anti-melanoma effect and the discussion about the obtained results was not clear and was not based on the real data that should be extracted from MTT assay. This manuscript could be considered to be published if the authors improve the biological experiments. The list of some important points to be revised is described below:

Title: Anti-melanoma effects are not appropriate to put in the title, since the authors only carried out cytotoxicity experiments, and there are no experiments in normal cell line, which is necessary to infer about the potential of extract and compounds as anti-melanoma drug. Besides, a large set of experiments, even in vitro, must be carried out to infer that compounds have anti-melanoma effects. I suggest removing it and change to “cytotoxicity effect”.

  1. 55-60: Why did the authors chose test antitumor effect of these compounds?

Fig. 1: Please edit the mass spectra to be possible to observe in a higher font the main m/z peaks. Besides, remove the upper blue information since they are not relevant to analyzing the spectra. Please insert in subtitle that each spectrum in figure A (or B) is a partial spectrum from… to…..

  1. 79: To analyze the total mass spectrum of crude extract it is not necessary to carry out the sequential UPLC-MS/MS. In my point of view, the spectra shown at Fig. 1 are not from UPLC-fractionated fraction.
  2. 95-96: I did not understand what is the mean of “The value was compared with different concentrations in each group (n=3)”. Please clarify
  3. 97: Which statistic test was used here?
  4. 170-171: I disagree to test the crude extract and each isolated compounds in the same concentration. The correct form to understand which compounds were responsible to observed activity would be tested each one in the correct proportion that they were in the mixture.
  5. 171-174: According to Fig. 6, compounds 2, 5, 6, 17, and 18 showed statistic cytotoxic effect. I did not understand why the authors said that all compounds inhibit cell survival. Besides, the sentence “However, the isolated compound had no obvious inhibitory effect” was referring to which compound? Regarding the used terms “inhibitory effect “ and “inhibited cell survival” I suggest using cytotoxic effect since the MTT assay evaluates it through mitochondrial dehydrogenases metabolism.
  6. 174-178: Where are showed these results? Why did the authors choose mix fractions 6, 7, 8, 9, and 11? Why did they mix 10,12,15,16,17,18?

There is no enough explanation about the cytotoxity effects. There are lacks that must be clarified and the authors must provide the results that they inserted in the text.

It is necessary to test the crude fraction and isolated compounds in normal cell line to provide evidence of less cytotoxicity in that cell line.    

  1. 183: What is the mean blank? Usually, we call it as control cells. The blank well is without cells and is useful to discount the background unspecific color in the colorimetric assay.
  2. 295- What were the used antibiotics? Which concentration of each one?
  3. 311: Which was the post-test used after ANOVA?

Author Response

Dear Reviewer,
Thank you for your suggestion, we had revised your comments.
We have attached the files reviewer#1-comments.
Thank you!

Reviewer 2 Report

This is a very interesting paper and a well-prepared one. Although phytochemical analysis and identification of potentially active compounds were major works for this paper, this reviewer still feels that authors should include supportive data on some pharmacological activity of compound 6 showing the highest value as below.

Authors should provide dose-dependent and time-dependent profiles of cytotoxicity by Compound 6.

Also, authors should present potential anti-cancer mechanism of compound 6 (eg., apoptosis-inducing activity, cell cycle arrest activity, necrosis effect., etc).

Molecular target should be also identified if possible.

Author Response

Dear Reviewer,
Thank you for your suggestion, we had revised your comments.
We have attached the files reviewer#2-comments.
Thank you!

Reviewer 3 Report

This paper is of particular scientific value primarily from the structural point of view, as providing a thorough analysis of P. taiwanensis extract including new, firstly described components. I have found several remarks, which I shall list in the order of the appearance in the text.

  1. In general, the paper may be supplemented with data obtained from human melanoma cell lines, Which would cost not much additional time.
  2. The data on melanization/melanogenesis markers are lacking. Melanin is the most important marker of melanoma, and the lack of melanin - an important feature of a melanoma. The molecular pathways of melanogenesis and neoplastic transformation overlap (partially) and affect one another, moreover, melanin may affect the results of colorimetric/spectrophoptometric assays, so that the melanization of the cells is an important observabla in studies such like this. Moreover, many components detected in the investigated extract are of aromatic/phenolic character, and may, again, affect the melanogenetic pathways being alternate substrates or inhibitors of the melanogenetic enzymes. If it is not possible now to provide one with detailed data on melanogenesis, please at least deliver macroscopic observations on pigmentation of the used cells.
  3. Line 38, it is good to refer here to the "melanoma triad" which gathers the most important symptoms of gaining the metastasis phenotype (asymmetry, white, blue and dark regions, a distinct net of thin and thick "veins" etc. Please mention (with proper citations)
  4. Line 48 - please mention the whole genus name (Pinus) when mentioning for the first time
  5. Line 53 - writing says
  6. (C-9) suggests
  7. Line 310 - of triplicates
  8. Line 170 - why cytotoxicity extract component 3 was not determined? If, indeed, it was not, please indicate it in the legend for Figure 6.

Author Response

Dear Reviewer,
Thank you for your suggestion, we had revised your comments.
We have attached the files reviewer#3-comments.
Thank you!

Reviewer 4 Report

The authors isolated and identified components from twigs of P. taiwnensis Hayata, and reported the results of evaluating the cytotoxicity of the isolated components in melanoma B16F10 cells. In the separation and identification stage of the components, a novel component that had not been previously reported was found, and the cytotoxic effect of the isolated components was also confirmed.

However, in order to further enhance the value of this study, it is recommended to add the following contents and experiments.

1. On page 7, lines 174-178, the authors describe the experimental results of fractions 4 and 5, but no figure data are available for the results.

2. The authors refer to the synergistic effect of the compounds in the cytotoxicity test on B16F10 melanoma cells. To verify this more clearly, It is recommended to conduct a cytotoxicity test of the crude extract on B16F10 cells.

Author Response

Dear Reviewer,
Thank you for your suggestion, we had revised your comments.
We have attached the files reviewer#4-comments.
Thank you!

Round 2

Reviewer 1 Report

Reviewer’s comments and questions which were not answered by the authors. They should be answered. 

  1. 170-171: I disagree to test the crude extract and each isolated compounds in the same concentration. The correct form to understand which compounds was responsible for the observed activity would be to test each one in the correct proportion that they were in the mixture.
  2. 171-174: According to Fig. 6, compounds 2, 5, 6, 17, and 18 showed statistic cytotoxic effect. I did not understand why the authors said that all compounds inhibit cell survival. Besides, the sentence “However, the isolated compound had no obvious inhibitory effect” was referring to which compound? Regarding the used terms “inhibitory effect “ and “inhibited cell survival” I suggest using cytotoxic effect since the MTT assay evaluates it through mitochondrial dehydrogenases metabolism.
  3. 174-178: Where are showed these results? Why did the authors choose mix fractions 6, 7, 8, 9, and 11? Why did they mix 10,12,15,16,17,18?

There is no enough explanation about the cytotoxicity effects. There are lacks that must be clarified and the authors must provide the results that they inserted in the text.

It is necessary to test the crude fraction and isolated compounds in normal cell line to provide evidence of less cytotoxicity in that cell line.    

  1. 183: What is the mean blank? Usually we call it as control cells. The blank well is without cells and is useful to discount the background unspecific color in the colorimetric assay.
  2. 295- What were the used antibiotics? Which concentration of each one?

Author Response

Dear Reviewer,
Thank you for your comments. 
We highlighted the changes in yellow in the manuscript, thank you for your comments.

Reviewer 2 Report

This paper is now acceptable.

Author Response

Dear Reviewer,
Thank you for the comments on our manuscript.
Thank you so much!

Reviewer 3 Report

All my suggestions have been addressed exhaustively enough to accept the present version of the manuscript.

Author Response

(The authors gave the same response as above.)

Reviewer 4 Report

The authors responded appropriately to the reviewer's suggestion. I think this manuscript deserves publication in this journal.

Author Response

(The authors gave the same response as above.)
